# 3D-ResNet-BiLSTM Model: A Deep Learning Model for County-Level Soybean Yield Prediction with Time-Series Sentinel-1, Sentinel-2 Imagery, and Daymet Data

Mahdiyeh Fathi [1] , Reza Shah-Hosseini [1,*] and Armin Moghimi [2]

1   School of Surveying and Geospatial Engineering, College of Engineering, University of Tehran, Tehran 14399-57131, Iran; mahdiyeh.fathi@ut.ac.ir
2   Ludwig-Franzius-Institute for Hydraulic, Estuarine and Coastal Engineering, Leibniz University Hannover, Nienburger Str. 4, 30167 Hannover, Germany; moghimi@lufi.uni-hannover.de
*   Correspondence: rshahhosseini@ut.ac.ir

**Abstract:** Ensuring food security in precision agriculture requires early prediction of soybean yield at various scales within the United States (U.S.), ranging from international to local levels. Accurate yield estimation is essential in preventing famine by providing insights into food availability during the growth season. Numerous deep learning (DL) algorithms have been developed to estimate soybean yield effectively using time-series remote sensing (RS) data to achieve this goal. However, the training data with short time spans can limit their ability to adapt to the dynamic and nuanced temporal changes in crop conditions. To address this challenge, we designed a 3D-ResNet-BiLSTM model to efficiently predict soybean yield at the county level across the U.S., even when using training data with shorter periods. We leveraged detailed Sentinel-2 imagery and Sentinel-1 SAR images to extract spectral bands, key vegetation indices (VIs), and VV and VH polarizations. Additionally, Daymet data was incorporated via Google Earth Engine (GEE) to enhance the model's input features. To process these inputs effectively, a dedicated 3D-ResNet architecture was designed to extract high-level features. These enriched features were then fed into a BiLSTM layer, enabling accurate prediction of soybean yield. To evaluate the efficacy of our model, its performance was compared with that of well-known models, including the Linear Regression (LR), Random Forest (RF), and 1D/2D/3D-ResNet models, as well as a 2D-CNN-LSTM model. The data from a short period (2019 to 2020) were used to train all models, while their accuracy was assessed using data from the year 2021. The experimental results showed that the proposed 3D-Resnet-BiLSTM model had a superior performance compared to the other models, achieving remarkable metrics ($R^2$ = 0.791, RMSE = 5.56 Bu Ac$^{-1}$, MAE = 4.35 Bu Ac$^{-1}$, MAPE = 9%, and RRMSE = 10.49%). Furthermore, the 3D-ResNet-BiLSTM model showed a 7% higher $R^2$ than the ResNet and RF models and an enhancement of 27% and 17% against the LR and 2D-CNN-LSTM models, respectively. The results highlighted our model's potential for accurate soybean yield predictions, supporting sustainable agriculture and food security.

**Keywords:** soybean; yield prediction; Conv3D; ResNet; BiLSTM; Sentinel 1–2; Daymet; Google Earth Engine (GEE)

## 1. Introduction

A high oil and protein content make soybeans a vital crop for food security. The United States (U.S.) is the leading global producer of this valuable commodity [1,2]. In 2021, the nation accomplished a historic feat by achieving a soybean production of 4.44 billion bushels (https://www.farmprogress.com/crops/farm-futures-survey-finds-record-2021-corn-crop, accessed on 15 January 2022). However, the soybean industry grapples with diverse challenges, from population growth and climate change [3]. Effectively addressing these challenges necessitates comprehensively evaluating crop type, soil quality, climate

conditions, environment, diseases, fertilizers, and seeds [4]. The U.S. Department of Agriculture (USDA) does not provide crop yield predictions until the subsequent March [5]. Therefore, early crop yield prediction becomes imperative in preventing famine by assessing food availability throughout the cultivation period. As such, timely and accurate crop yield prediction is paramount in evaluating trade balances, enhancing food security, formulating production, storage, and transportation strategies, and facilitating urbanization [6].

Accurate crop yield prediction relies on two primary techniques: traditional ground observations and advanced Remote Sensing (RS). Traditional methods are highly accurate but are costly and time-consuming, limiting their feasibility for large-scale applications like state-level assessments [4]. In recent years, RS technology has gained popularity for crop yield prediction. Its advantages include large-scale coverage, continuous monitoring, multispectral capabilities, affordability, and long-term data archiving across various spatial, spectral, and temporal resolutions [4,7]. Furthermore, the rich multispectral data within RS images allows us the opportunity to derive a wide range of valuable Vegetation Indices (VIs) (e.g., Normalized Vegetation Index (NDVI), Enhanced Vegetation Index (EVI), and Green Normalized Vegetation Index (GNDVI)). These indices can be utilized to monitor the phenology and growth of crops. Soil attributes such as pH, type, and moisture, coupled with features like Land Surface Temperature (LST), integrated drought indices, precipitation, vapor pressure, and humidity, have also been employed for crop yield prediction [8,9].

The prediction of crop yield data can be achieved via two main categories of models: process-based biophysical (PB) and machine-learning (ML) models. PB models (e.g., Agricultural Production System Simulator (APSIM), Decision Support System for Agro-Technology Transfer (DSSAT)) dynamically simulate crop yield by employing well-calibrated crop growth models. This framework often uses RS data to reinitialize, recalibrate, or update state variables in a model at a higher spatial resolution than the driving data. Nevertheless, calibrating process-based models at larger scales remains challenging, requiring various field measurements [1,10]. Consequently, in numerous scenarios requiring cost-effectiveness and flexible modeling of intricate patterns, ML-based algorithms are often preferred [11]. Traditional ML models such as Support Vector Machine (SVM) and Random Forest (RF) have proven effective in crop yield prediction [12–14]. However, these algorithms might struggle to extract advanced features from the input data. This limitation has driven the adoption of Deep Learning (DL) methods, including Convolutional Neural Networks (CNN), Recurrent Neural Networks (RNN), and the specific Long Short-Term Memory (LSTM) architecture. These models can extract intricate features from basic ones and effectively represent the complex correlations between the input and output variables using multiple hidden layers [4,15]. Therefore, these models have been widely developed for soybean yield prediction models in recent years. For example, You et al. [16] combined a Gaussian Process component to a CNN or LSTM for predicting crop yield using MODIS Land Surface Temperature (LST) data and MODIS Surface Reflectance (SR) data between 2003 and 2015 in the U.S. Sun et al. [5] also introduced an innovative deep CNN-LSTM to predict soybean yield at the county level within the U.S. from 2003 to 2015 using weather and MODIS LST and SR datasets. Similarly, Terliksiz et al. [17] designed a 3D-CNN model for soybean yield prediction in Lauderdale County using MODIS LST and SR data between 2003 and 2016. Khaki et al. [18] developed a CNN-RNN model that effectively captured the temporal relationships between environmental factors and the genetic enhancement of seeds without requiring access to genotype data. They used yield performance, management, weather data, and soil data variables to predict corn and soybean yield between 1980 and 2018 in the U.S. Khaki et al. [19] also developed the Yield-Net model, which utilized MODIS products, including MOD09A1 and MYD11A2, from 2004 to 2018 to predict crop yields. Schwalbert et al. [20] also designed an LSTM model to forecast soybean yield using VIs like NDVI, EVI, LST, and precipitation during southern Brazil's growing season between 2003 and 2016. Zhu et al. [21] introduced a DL-based Adaptive Crop Model (DACM) for accurate soybean yield prediction in the U.S. between 2003 and 2017, using MODIS LST data and MODIS SR data.

While the previously mentioned studies have shown commendable progress and promising results in soybean yield estimation, specific challenges persist. The MODIS has been extensively employed for soybean yield prediction due to its high temporal resolution [5,7,9,17], but its accuracy is limited by its low spatial resolution. However, the potential of even higher-resolution images, such as those from Sentinel-2, which provide rich spectral information including red-edge bands, needs more attention. Additionally, the potential of combining Sentinel-2 and Sentinel-1 images, along with weather and climatology variables, to improve prediction accuracy has been less regarded. Furthermore, current approaches often employ 1D/2D-CNN-LSTM models to predict crop yield, limiting their ability to incorporate future data and demanding substantial computational resources [5,22,23]. While these models have demonstrated robust prediction abilities with long-time span training data, assessing their performance in scenarios where the data is limited to a shorter period is imperative.

In response to these challenges, we introduced the 3D-ResNet-BiLSTM model as a solution for early and accurate county-level soybean yield prediction for the U.S. during the growing season, using a short period dataset derived from the Sentinel 1, Sentinel 2, and Daymet data. Employing the 3D-ResNet architecture in our model allows us to capture rich spatial features from the input data, facilitating enhanced feature extraction and improved performance in yield prediction. A notable advantage of the 3D-ResNets is their incorporation of residual blocks, enabling the network to learn residual functions that streamline deep network training [24]. The predictive aspect of our model is powered via the Bidirectional LSTM (BiLSTM) module, enabling bidirectional data utilization during calculations. This bidirectional processing is particularly advantageous for sequential data, incorporating both preceding and subsequent information, resulting in heightened prediction accuracy [22]. Moreover, our proposed method evaluates soybean yield prediction specifically during the growing season, providing valuable insights into temporal variability and challenges, which has received comparatively less attention in the prior literature.

The remainder of this study is structured as follows: Section 2 provides in-depth details on the study area, datasets, methodology, the 3D-ResNet-BiLSTM model, and the evaluation metrics. Section 3 presents the experimental results, while Section 4 describes an extensive discussion that contextualizes the results of the experiments. Finally, Section 5 presents concluding remarks and overall conclusions.

## 2. Materials and Methods

### 2.1. Study Area

The study was located in the U.S., including eighteen states: North Dakota, South Dakota, Nebraska, Kansas, Oklahoma, Minnesota, Iowa, Missouri, Arkansas, Louisiana, Wisconsin, Illinois, Michigan, Indiana, Ohio, Kentucky, Tennessee, and Mississippi (see Figure 1). The research was carried out from 2019 to 2021, centering on the growth of soybeans, a key cereal crop cultivated within the study area. Soybeans are commonly sown between May and early June, with harvesting in the late months of September and October (https://www.ers.usda.gov/topics/crops/soybeans-and-oil-crops/oil-crops-sector-at-a-glance/, accessed on 1 September 2021).

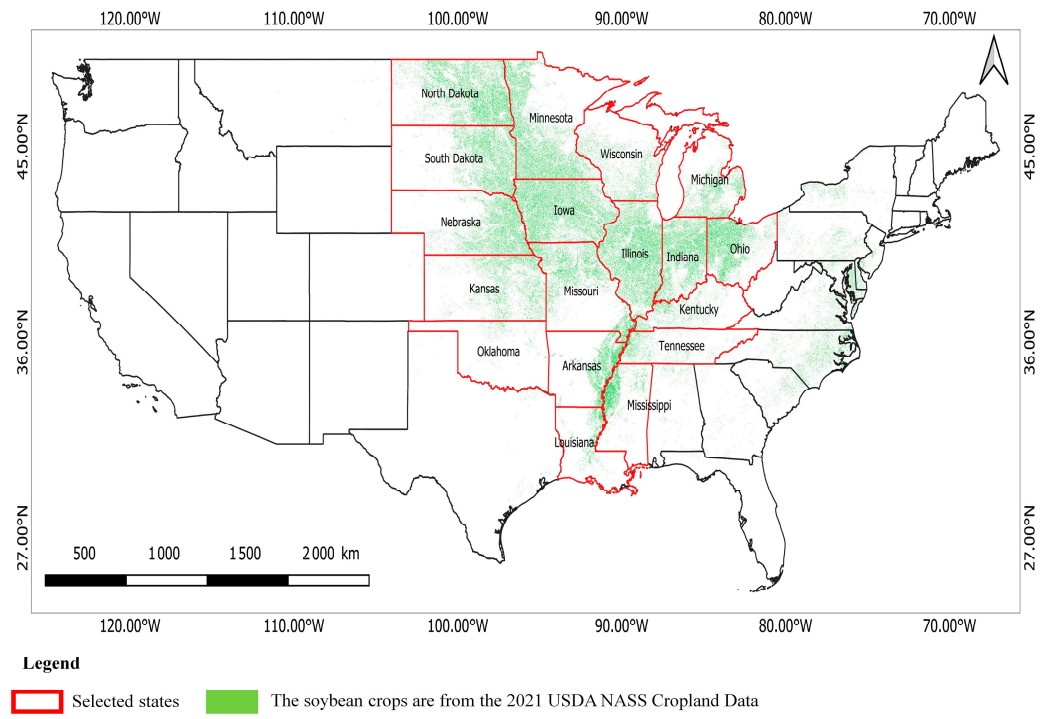

**Legend**

☐ Selected states  ■ The soybean crops are from the 2021 USDA NASS Cropland Data

**Figure 1.** Study area: U.S. states outlined in red indicate the specific focus for county-level soybean yield estimation. The soybean crops displayed are from the 2021 USDA NASS Cropland Data Layer.

### 2.2. Dataset

This study employed a variety of data sources to predict soybean yield, including Sentinel 1 SAR (COPERNICUS/S1_GRD), Sentinel-2 Surface Reflectance (S2_SR_HARMONIZED), Daymet weather (Daymet V4), USDA Yield, Crop Land Data Layer (CDL), and County Boundaries data.

Sentinel-1 collects data from a dual-polarization C-band Synthetic Aperture Radar (SAR) instrument at 5.405GHz, with each scene including 1 or 2 polarization bands out of four possible options. The available combinations are single-band VV or HH and dual-band VV + VH or HH + HV, with a pixel size of 10 m [25].

Sentinel-2 provides high-resolution, multi-spectral imagery for monitoring vegetation, soil, water cover, and more, with a pixel size of 10, 20, and 60 m [25].

The Daymet data provides highly accurate and detailed gridded estimates of daily weather parameters across Continental North America, Hawaii, and Puerto Rico, with a resolution of 1 km × 1 km. This invaluable resource offers unparalleled insights into these regions' weather patterns and conditions, allowing for more precise planning and decision making in various fields [26]. The Crop Land Data Layer (CDL) with a spatial resolution of 30 m was retrieved from the USDA, which employs the Decision Tree approach to categorize agricultural areas using various sensors [27]. Non-soybean pixels were masked using CDL.

The USDA creates an annual report outlining crop acreage, yields, areas harvested, and other production information (https://quickstats.nass.usda.gov/, accessed on 15 January 2021). The data from Sentinel 1, Sentinel 2, and Daymet were all retrieved via the Google Earth Engine (GEE) cloud-based platform [28]. Training and test data were gathered within the timeframe of 2019 to 2021. Cloud-covered and non-soybean pixels were excluded to compute specific features. These selected features were then employed as inputs for DL models, enabling the prediction of soybean yield. Table 1 displays the statistical characteristics of yield observations for both the training and test datasets.

**Table 1.** Sample plot yield statistics for the year in the study area.

| Type | Year | Number of Samples | Min (Bu AC$^{-1}$) | Max (Bu AC$^{-1}$) | Mean (Bu AC$^{-1}$) | Std. (Bu AC$^{-1}$) |
|---|---|---|---|---|---|---|
| train | 2019 | 437 | 21.80 | 65.50 | 49.64 | 8.15 |
| train | 2020 | 682 | 24.70 | 72.30 | 52.37 | 8.58 |
| test | 2021 | 601 | 13.80 | 77.30 | 53.25 | 12.20 |

### 2.3. Methodology

This methodology is designed to predict soybean yields at the county level in the U.S. during the in-season period, explicitly focusing on August and utilizing the 3D-ResNet-BiLSTM model. As depicted in Figure 2, the approach involves two fundamental steps. Initially, relevant features are extracted from the Sentinel-1, Sentinel-2, and Daymet data within the GEE platform, resulting in 23 distinct features spanning 2019, 2020, and 2021. These features serve as the independent variables for the model. Correspondingly, the USDA soybean yield data are considered dependent variables, forming the input data for constructing the 3D-ResNet-BiLSTM model.

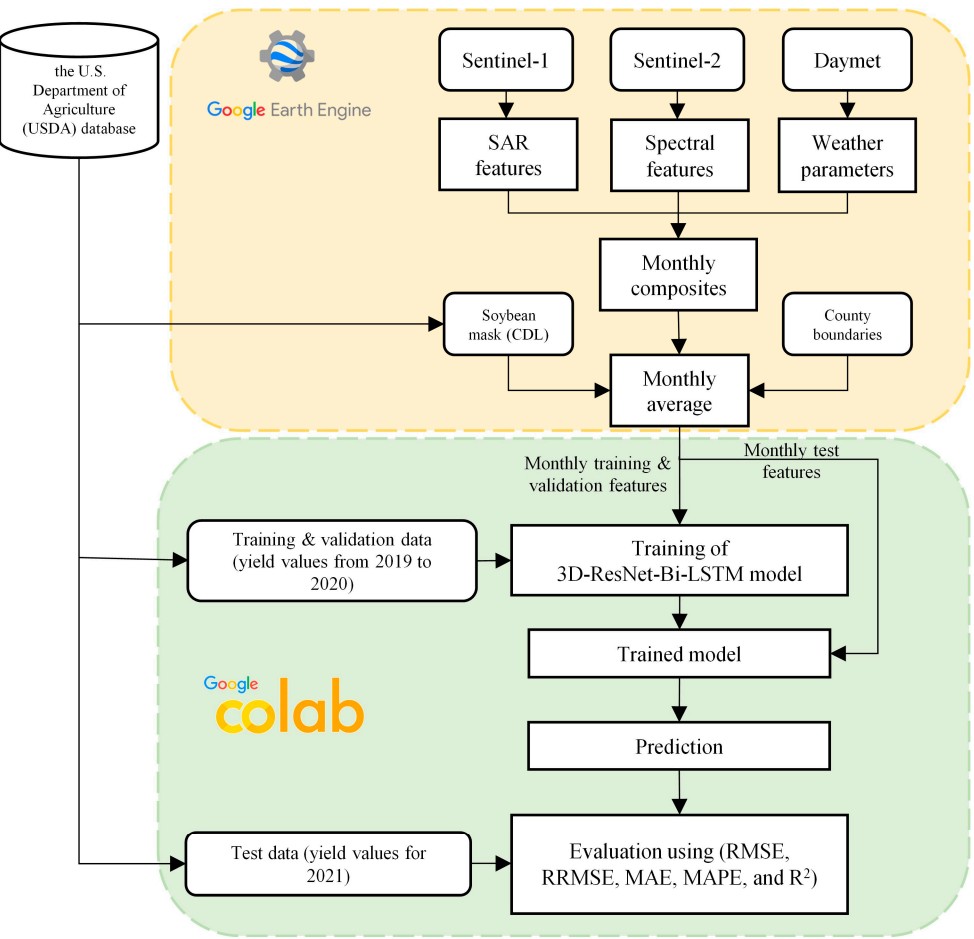

**Figure 2.** Workflow of the proposed method.

The input dataset is categorized into training, validation, and test datasets, and the model is trained using data from 2019 and 2020. The input dataset is classified into training, validation, and test datasets, and the model is trained using data from 2019 and 2020. The trained model is then utilized to predict soybean yields based on the test feature vector for the year 2021. Using USDA's test yield value data from 2021, model predictions are

subsequently evaluated. A detailed exposition of the feature extraction and 3D-ResNet-BiLSTM model is provided in Sections 2.3.1 and 2.3.2.

### 2.3.1. Feature Selection

In this study, crop yield estimation was facilitated using various RS features. Sentinel-2 SR data were employed to derive various suitable Vis (see Table 2) such as DVI, GNDVI, EVI, LSWI, RVI, SAVI, VARIGREEN, WDRVI, and NDVI, drawing from established works (see Table 2). Furthermore, the predictive power was improved by tapping into the distinct spectral bands of Sentinel-2 data, including Blue, Red, Green, Near Infrared (NIR), narrow NIR (nNIR), Red Edge 1/2/3, and Shortwave Infrared (SWIR) 1/2. In addition, the study incorporated Sentinel-1 SAR polarization VV and VH data alongside weather-derived features from Daymet, like precipitation and vapor pressure. This comprehensive feature set was integrated within the GEE cloud-based platform. The feature generation process for each county within the GEE system included four key steps: (1) the creation of monthly composites, (2) masking out cloud-covered regions, (3) excluding non-soybean areas using the Cropland Data Layer (CDL), and (4) the calculation of the monthly feature averages for soybean fields within each county, delineated by county boundaries. The temporal progression of the extracted features for soybean fields during the planting season is visually depicted in Figure 3.

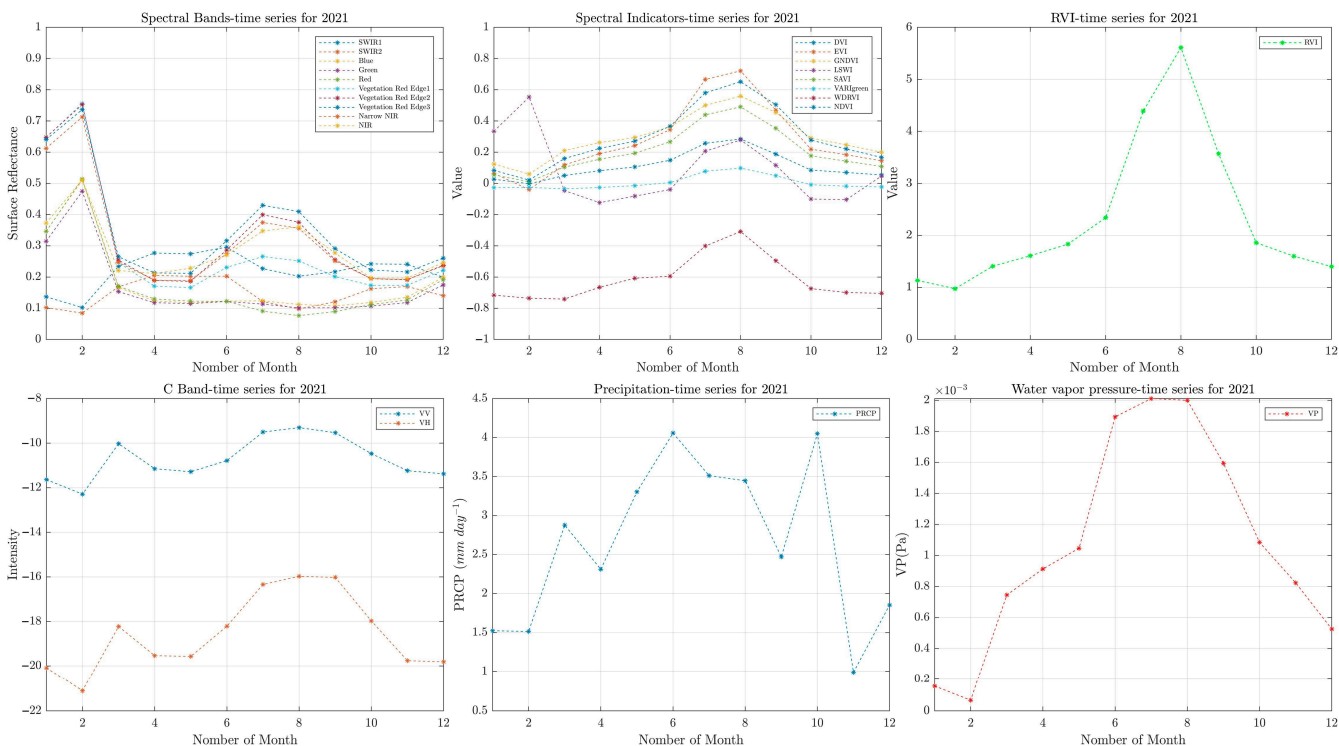

**Figure 3.** The time series curve of the extracted features for soybean fields during planting.

**Table 2.** The extracted Indicators from Sentinel 1 and Sentinel 2.

| Name | Formula | Ref. |
|---|---|---|
| Normalized Difference Vegetation Index (NDVI) | $\dfrac{\rho_{Nir} - \rho_{Red}}{\rho_{Nir} + \rho_{Red}}$ | [29] |
| Wide Dynamic Range Vegetation Index (WDRVI) | $\dfrac{0.1 \times \rho_{Nir} - \rho_{Red}}{0.1 \times \rho_{Nir} + \rho_{Red}}$ | [30] |
| Enhanced Vegetation Index (EVI) | $\dfrac{2.5 \times (\rho_{Nir} - \rho_{Red})}{(\rho_{Nir} + 6 \times \rho_{Red} - 7.5 \times \rho_{Blue} + 1)}$ | [31] |

**Table 2.** *Cont.*

| Name | Formula | Ref. |
|---|:---:|:---:|
| Difference Vegetation Index (DVI) | $\rho_{Nir} - \rho_{Red}$ | [32] |
| Land Surface Water Index (LSWI) | $\dfrac{\rho_{Nir} - \rho_{Swir}}{\rho_{Nir} + \rho_{Swir}}$ | [33] |
| Ratio Vegetation Index (RVI) | $\dfrac{\rho_{Nir}}{\rho_{Red}}$ | [34] |
| Visible Atmospherically Resistant Index Green (VARIgreen) | $\dfrac{\rho_{Green} - \rho_{Red}}{\rho_{Green} + \rho_{Red} - \rho_{Blue}}$ | [35] |
| Soil Adjusted Vegetation Index (SAVI) | $\dfrac{\rho_{Nir} - \rho_{Red}}{\rho_{Nir} + \rho_{Red} + 0.5} \times 1.5$ | [36] |
| Green Normalized Difference Vegetation Index (GNDVI) | $\dfrac{\rho_{Nir} - \rho_{Green}}{\rho_{Nir} + \rho_{Green}}$ | [30] |

### 2.3.2. 3D-ResNet-BiLSTM Model Architecture

The proposed 3D-ResNet-BiLSTM model is a hybrid architecture that combines the 3D-ResNet and BiLSTM, as illustrated in Figure 4. The 3D-ResNet model is initially employed to extract high-level features from the input data previously generated from selected features. Subsequently, the BiLSTM algorithm is utilized to predict soybean yield based on these extracted features. By merging these two components, our model effectively captures intricate relationships between the input RS data and the in situ crop yield, resulting in more accurate predictions.

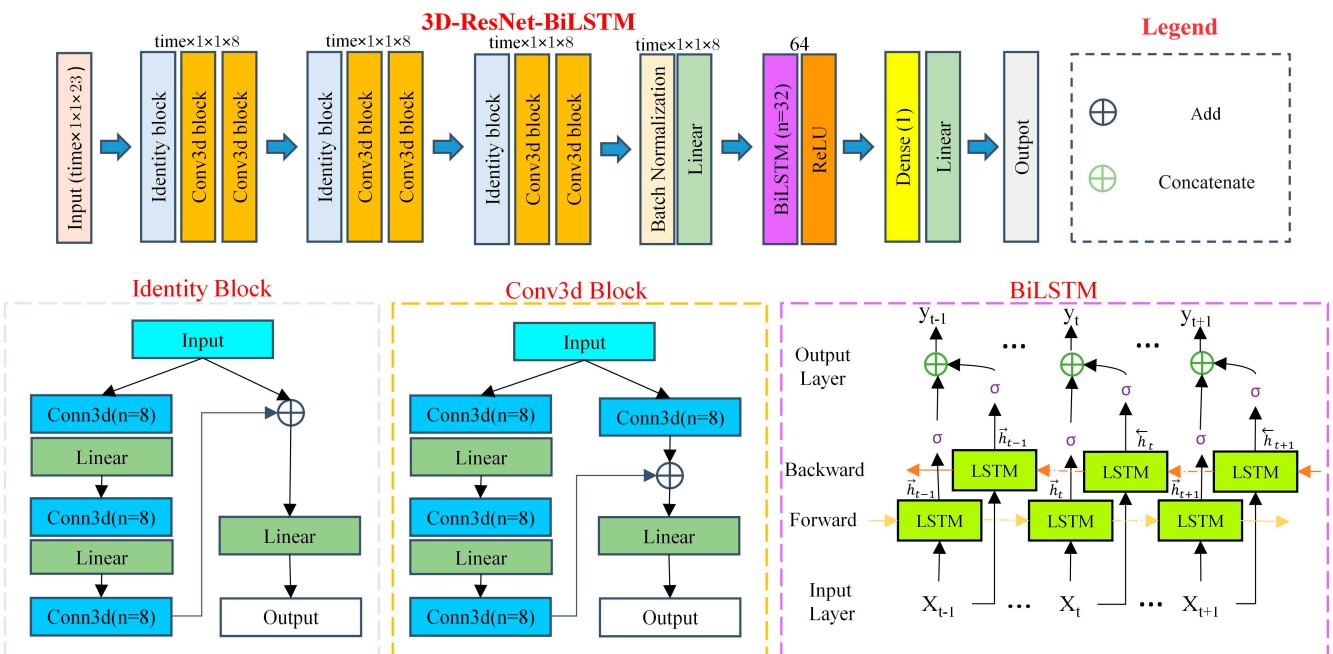

**Figure 4.** The architecture of the proposed 3D-ResNet-BiLSTM model.

### 3D-ResNet Component

Our 3D-ResNet component was designed to handle spatial and temporal factors within SAR, optical, and weather data, ensuring highly accurate soybean crop yield estimations. This design captures crop growth trends and their spatial distribution in fields, as illustrated in Figure 4. The 3D-ResNet consists of three layers, each comprising an Identity block and two Conv3D blocks, custom-tailored to the unique dynamics of soybean crops. Each preceding block's output is the subsequent input in this cascading design.

The Identity block assumes a central role within this framework, featuring a sequence of 3D convolutional layers and a skip connection block. These skip connections preserve

unique SAR, vIs, and weather data attributes, facilitating gradient flow in multi-modal data and enhancing soybean crop estimation [24]. This aspect is crucial in soybean crop estimation, enabling the model to learn and capture intricate relationships between the input features and yield outcomes.

The Conv3D block includes a set of 3D CNNs, enhancing the model's capacity to analyze spatial and temporal information within SAR, optical VIs, and weather features. Similarly, the Conv3D block, equipped with skip connections, captures spatiotemporal dynamics, which is essential for accurate crop yield estimation [24]. This capacity is precious as it reveals the interplay between the temporal trends and spatial arrangements, providing critical insights into crop development and eventual yield outcomes.

The input $X$ of the identity-block input passes through a sequence of operations: 3D convolutional layer > linear activation function > 3D convolutional layer > linear activation function > 3D convolutional layer, resulting in the extraction of features $F_{IB}$. These extracted features, $F_{IB}$, are added to $X$ and then processed via a linear activation function, denoted as $f_L$, which serves as the input for the subsequent Conv3D block (Equations (1) and (2)).

$$F_{IB} = F_{IB} + X \tag{1}$$

$$F_{IB} = f_L(F_{IB}) \tag{2}$$

The input $F_{IB}$ is then processed via the Conv3D block, which involves a 3D convolutional layer to extract features $F_X$, as given by:

$$F_X = W F_{IB} + b \tag{3}$$

where $W$ represents the weight matrix and b is the bias term.

Moreover, within the Conv3D block, the input $F_{IB}$ undergoes a series of operations: 3D convolutional layer > linear activation function > 3D convolutional layer > linear activation function > 3D convolutional layer, to extract the $F_{cB}$ features. These features are further added to $F_X$ and passed through the function $f_L$ to construct the input for the next block, as described by:

$$F_o = F_{cB} + F_X \tag{4}$$

$$F_o = f_L(F_o) \tag{5}$$

Bi-LSTM Component

Following the feature extraction via the 3D-ResNet, the data undergoes Batch Normalization with linear activation and then enters the Bi-LSTM layer with ReLU activation. This configuration enables precise soybean yield prediction, benefiting from the reverse-order hidden state set for context capture [22].

In this way, a BiLSTM cell is initially fed with an input sequence, $x = (x_1, x_2, \ldots, x_n)$, where $n$ represents the length of the sequence. Furthermore, $\overrightarrow{H}$ denotes the forward hidden sequence, $\overleftarrow{H}$ means the backward hidden sequence, and $y_t = (y_1, y_2, \ldots, y_n)$ is the output sequence. The final encoded output vector is the combined effect of both the forward and backward information flow, i.e., $y_t = f\left(\overrightarrow{H}, \overleftarrow{H}\right)$. The mathematical framework of the BiLSTM neural networks' architecture is presented in Equations (6)–(8) [37]

$$\overrightarrow{H} = \sigma(w_{\overrightarrow{h}x} x_t + w_{\overrightarrow{h}\overrightarrow{h}} xh_t + b_{\overrightarrow{h}}) \tag{6}$$

$$\overleftarrow{H} = \sigma\left(w_{\overleftarrow{h}x} x_t + w_{\overleftarrow{h}\overleftarrow{h}} xh_t + b_{\overleftarrow{h}}\right) \tag{7}$$

$$y_t = w_{y\overrightarrow{h}} \overrightarrow{h}_t + w_{y\overleftarrow{h}} \overleftarrow{h}_t + b_y \tag{8}$$

where $\sigma$ represents the sigmoid activation function, mapping values to the [0, 1] range. Finally, a dense layer with a linear activation function is applied to the output of BiLSTM to predict soybean yield.

### 2.4. Evaluation Metrics

The performance of the proposed and considered models was evaluated using some metrics, including Root Mean Square Error (RMSE), Mean Absolute Error (MAE), Mean Absolute Percentage Error (MAPE), Relative Root Mean Squared Error (RRMSE), and Coefficient of determination ($R^2$), which can be calculated as follows [38,39]:

$$RMSE = \sqrt{\sum_{i=1}^{N} \frac{\left(y_{pred}^i - y_{obs}^i\right)^2}{N}} \tag{9}$$

$$\text{MAE} = \frac{1}{N}\sum_{i=1}^{N}\left|y_{pred}^i - y_{obs}^i\right| \tag{10}$$

$$MAPE = \frac{1}{N}\sum_{i=1}^{N}\frac{\left|y_{pred}^i - y_{obs}^i\right|}{y_{obs}^i} \tag{11}$$

$$R^2 = 1 - \frac{\sum_{i=1}^{N}\left(y_{pred}^i - y_{obs}^i\right)^2}{\sum_{i=1}^{N}\left(y_{obs}^i - y_{mean}\right)^2} \tag{12}$$

$$RRMSE = \frac{RMSE}{y_{mean}} \times 100 \tag{13}$$

where $N$ is the number of the test samples, $y_{obs}^i$ and $y_{pred}^i$, respectively, are the observed and predicted data $i$th test samples, and $y_{mean}$ represents the average of the observed data.

## 3. Experimental Results

### 3.1. Experimental Setup

All the experiments were conducted using the RS data extracted and prepared within GEE. The experiments were implemented using a Python script in Google Colaboratory (Colab), utilizing a TPU and 12 GB of RAM. As previously discussed, the proposed model architecture incorporated 23 features extracted from the Sentinel 1–2 and Daymet data as inputs. Accordingly, our model utilized input tensors with dimensions of $8 \times 1 \times 1 \times 23$ (time steps $\times$ features) and $9 \times 1 \times 1 \times 23$ (time steps $\times$ features) for the respective months of August and September, specifically for the in-season growth period. To compare the proposed model's performance, we evaluated it against 1D/2D/3D-ResNet, ResNet, 2D-CNN-LSTM [5], RF, and LR. We also designed a 3D-RsNet architecture by removing the BiLSTM layer from the proposed architecture. The 1D/2D-ResNet architectures were implemented by replacing the Conv1D/2D layer with a Conv3D layer in the 3D-ResNet architecture. The dense layer was replaced with a Conv3D layer in the 3D-ResNet architecture to form the ResNet architecture. The training phase of all models employed is based on the MAPE as the loss function, coupled with the Adam optimizer set at a uniform learning rate of 1.10. The number of parameters used in the models under consideration is listed in Table 3.

As demonstrated in Table 3, ResNet-based models outperformed the 2D-CNN-LSTM model regarding computational efficiency, making them a more efficient choice for crop yield estimation. The loss curves using training and validation datasets for the proposed method and all considered models have been demonstrated in Figure 5.

**Table 3.** The number of parameters and run time for all of the considered models in soybean yield prediction.

| | Aug. | | Sept. | |
| --- | --- | --- | --- | --- |
| **Model** | **Parameter** | **Time** | **Parameters** | **Time** |
| 3D-ResNet-BiLSTM | 12,929 | 07 min 25 s | 12,929 | 07 min 59 s |
| 3D-ResNet | 2433 | 06 min 39 s | 2441 | 06 min 56 s |
| 2D-ResNet | 2433 | 05 min 05 s | 2441 | 05 min 20 s |
| 1D-ResNet | 2433 | 05 min 05 s | 2441 | 05 min 09 s |
| ResNet | 4505 | 03 min 49 s | 4809 | 03 min 48 s |
| 2D-CNN-LSTM | 372,353 | 15 min 21 s | 375,745 | 18 min 53 s |

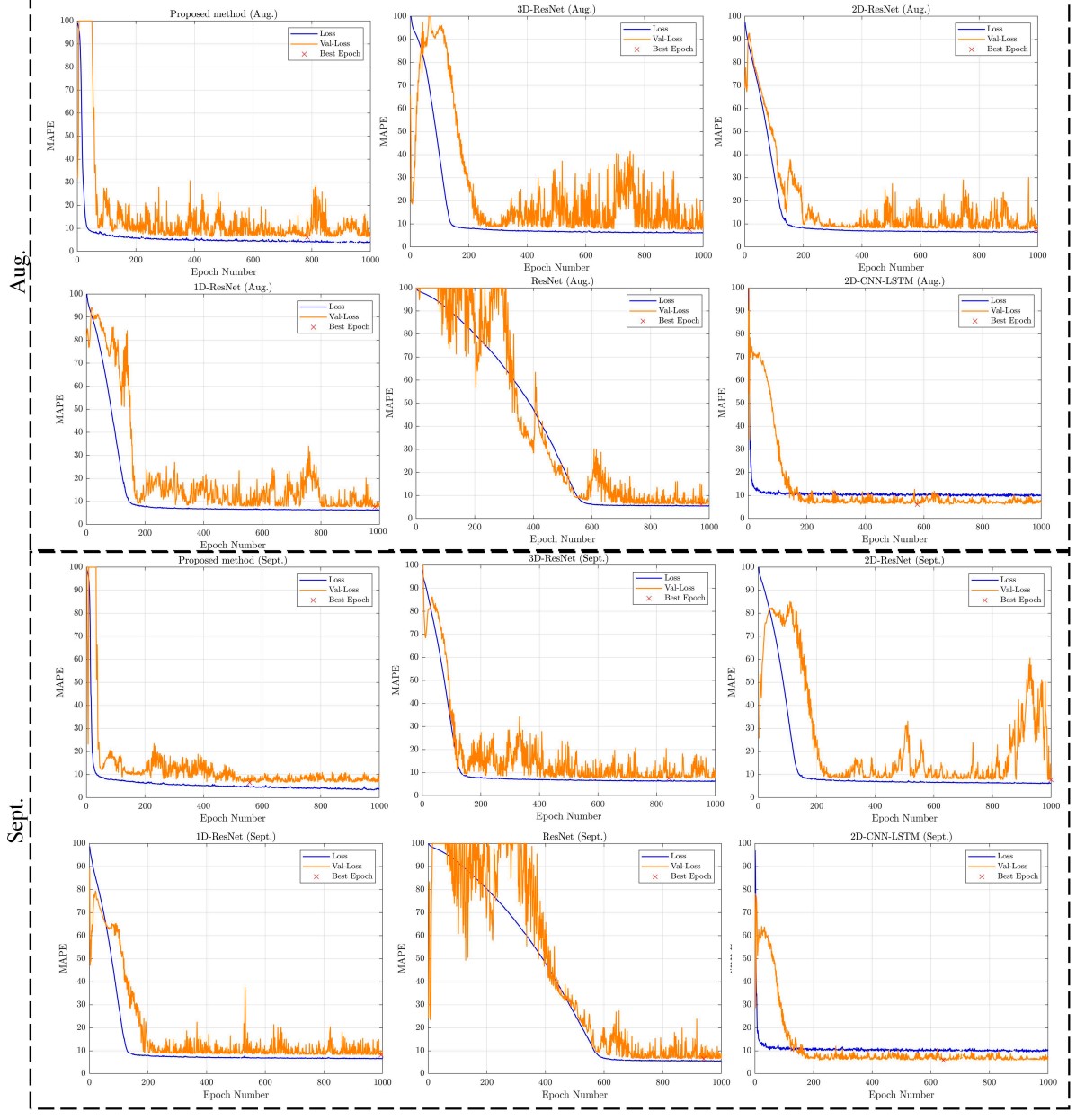

**Figure 5.** Loss curves using proposed and considered models on training and validation datasets.

As seen in Figure 5, the proposed 3D-ResNet-BiLSTM model exhibited a rapid and substantial reduction in its loss values, markedly diverging from the ResNet model, which displayed a more gradual decline. This observed difference highlights the unique impact of integrating Conv3D and BiLSTM networks into our architectural framework. Examining the validation loss curves revealed higher fluctuations in August compared to September. Moreover, these curves illustrated that including the extracted features in subsequent months reduced the range of fluctuations. The evaluation of validation loss curves for the 2D-CNN-LSTM did not indicate over/underfitting and a marginal improvement in results was possible by extending the number of epochs. However, for the validation loss curves of 2D-ResNet, overfitting was evident between epochs 800 and 1000, marked by a substantial increase in the gap between the validation loss and the training loss. In a broader context, preserving the best model based on loss validation might be more prudent for yield prediction, considering the potential scenario where the model fails to achieve proper convergence with the inclusion of validation data [40]. Our model architecture demonstrated a superior performance to the models evaluated in scenarios with short-period training data.

### 3.2. Comparative Results of the Soybean Yield Prediction

In this subsection, we presented comparative results for our model and another model under consideration. The results for both the proposed and considered models in predicting soybean yield are displayed in Table 4, covering the growth period during August and September 2021.

**Table 4.** Performance of proposed and considered models for soybean yield prediction during the Growing In-Season period (i.e., August and September).

| Aug. | | | | | |
|---|---|---|---|---|---|
| **Model** | **RMSE (Bu Ac$^{-1}$)** | **R$^2$** | **MAE (Bu Ac$^{-1}$)** | **MAPE (%)** | **RRMSE (%)** |
| 3D-ResNet-BiLSTM | 5.53 | 0.79 | 4.28 | 8.80 | 10.38 |
| 3D-ResNet | 5.71 | 0.78 | 4.50 | 9.41 | 10.72 |
| 2D-ResNet | 6.03 | 0.75 | 4.85 | 10.13 | 11.32 |
| 1D-ResNet | 6.12 | 0.74 | 4.96 | 10.45 | 11.49 |
| ResNet | 6.34 | 0.73 | 5.23 | 10.99 | 11.90 |
| 2D-CNN-LSTM | 7.61 | 0.61 | 6.05 | 12.64 | 14.29 |
| RF | 6.56 | 0.71 | 5.44 | 11.22 | 12.31 |
| LR | 7.55 | 0.61 | 5.73 | 10.77 | 14.10 |
| Sep. | | | | | |
| **Model** | **RMSE (Bu Ac$^{-1}$)** | **R$^2$** | **MAE (Bu Ac$^{-1}$)** | **MAPE (%)** | **RRMSE (%)** |
| 3D-ResNet-BiLSTM | 5.60 | 0.79 | 4.42 | 9.21 | 10.61 |
| 3D-ResNet | 5.72 | 0.78 | 4.48 | 9.43 | 10.74 |
| 2D-ResNet | 5.95 | 0.76 | 4.65 | 9.72 | 11.17 |
| 1D-ResNet | 6.05 | 0.75 | 4.83 | 10.19 | 11.36 |
| ResNet | 6.65 | 0.70 | 5.50 | 11.74 | 12.48 |
| 2D-CNN-LSTM | 7.79 | 0.59 | 6.40 | 13.57 | 14.62 |
| RF | 6.59 | 0.71 | 5.44 | 11.23 | 12.37 |
| LR | 9.58 | 0.38 | 7.32 | 13.06 | 17.99 |

The results from Table 4 indicated that the 3D-ResNet-BiLSTM model achieved the best performance, demonstrating its capability for predicting soybean yield using multi-sensor RS data. This model accurately forecasted soybean yield, especially in August, before the harvest season. For instance, the RMSE of the proposed model was 5.53 Bu Ac$^{-1}$

in August and 5.60 Bu Ac$^{-1}$ in September, marking an improvement of about 3% and 30% compared to the 3D-ResNet (the second-best model) and the LR (the worst model), respectively. Moreover, the 3D-ResNet-BiLSTM model, with an RRMSE of approximately 10.5% and an R$^2$ of 0.79, emerged as the most accurate soybean yield predictor, closely followed by the 3D-ResNet. This improvement could be attributed to the incorporation of temporal insights complementing the spatial information provided via the 3D architecture.

To better understand our proposed model's effectiveness, we generated error maps for August and September using our model and the models under consideration, as depicted in Figures 6 and 7.

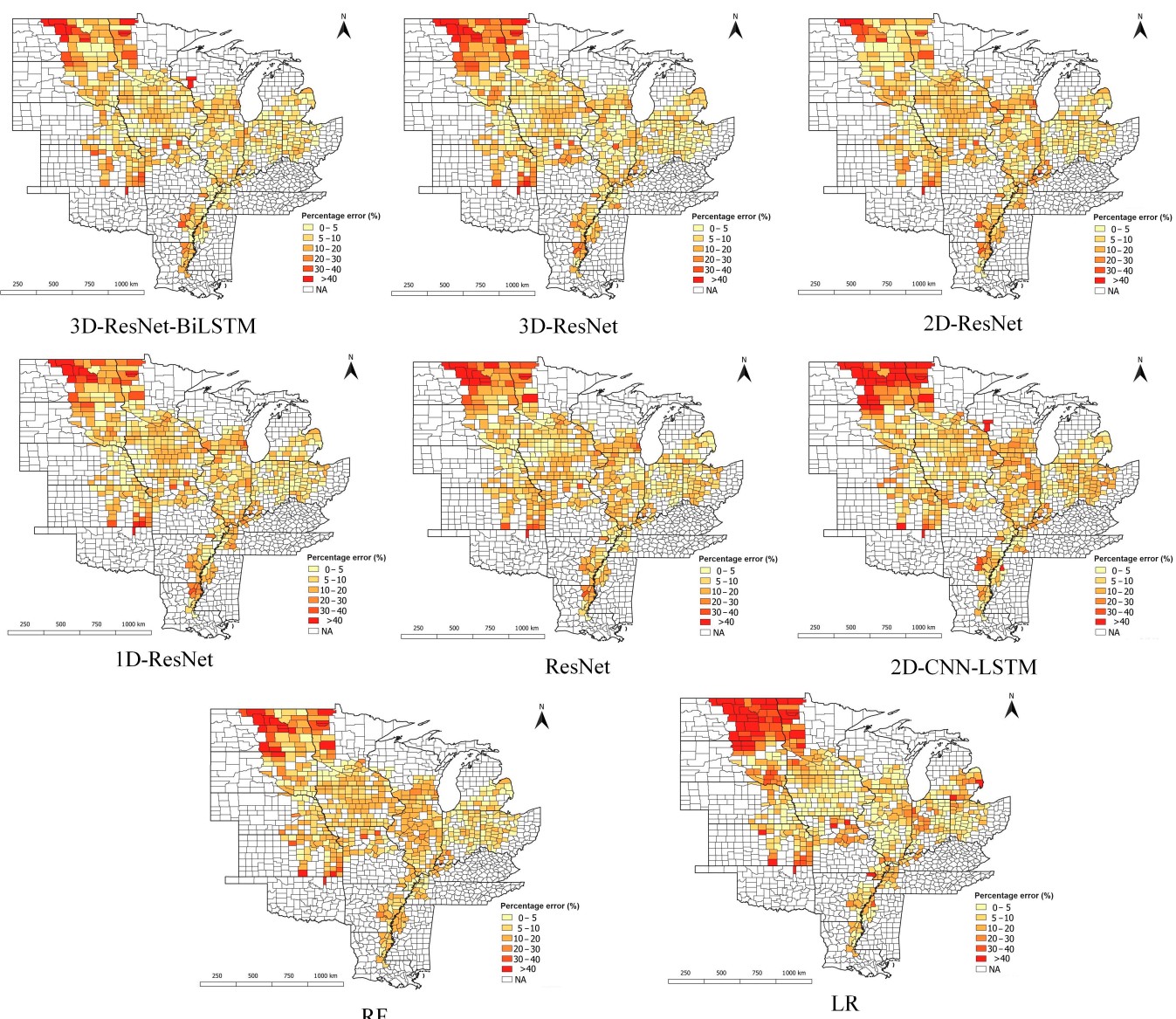

**Figure 6.** Error maps generated using both the proposed and considered models in August.

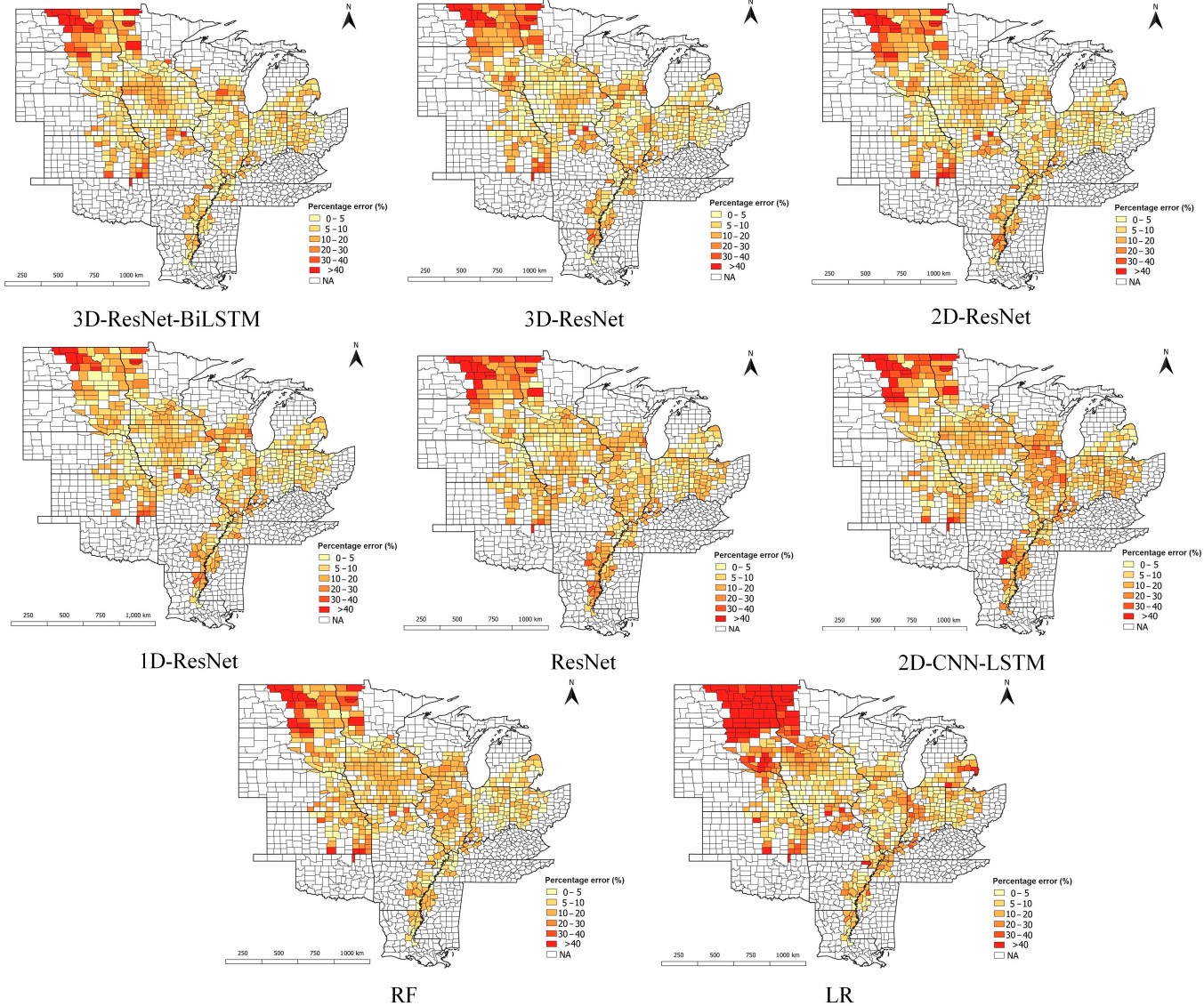

**Figure 7.** Error maps generated using both the proposed and considered models in September.

As observed in Figures 6 and 7, the combined operation of the ResNet network, Conv3d network, and BiLSTM network simultaneously reduced errors and rendered the error maps brighter. The error maps also revealed that counties with lower yields also tended to have higher percentage errors, represented by darker colors on the maps. Several factors could reduce soybean yield, including climate changes, fertilization, irrigation, drought, soil characteristics, disease, and pests. Notably, Oklahoma State had the highest MAPE (132.51%) due to a lack of training data in that particular study. The 2D-CNN-LSTM and LR models exhibited poor alignment between the predicted and observed yield values.

Figure 8 depicts scatter plots between the predicted and observed yields for the proposed and considered models. These scatter plots confirmed the superior performance of the 3D-ResNet-BiLSTM model in yield prediction when using a combination of the Sentinel-1, Sentinel-2, and Daymet data as inputs.

The scatter plots clearly showed lower RMSE and RRMSE values and higher $R^2$ values, indicating a more robust and more accurate relationship between the predicted and observed soybean yield values. Furthermore, using the 3D-ResNet-BiLSTM architecture was notably more effective in improving the accuracy of soybean yield prediction compared to 1D/2D/3D-ResNet, ResNet, 2D-CNN-LSTM, RF, and LR. This highlights the advantage

of using feature extraction with 3D-ResNet and yield prediction with BiLSTM, mainly when dealing with limited training samples.

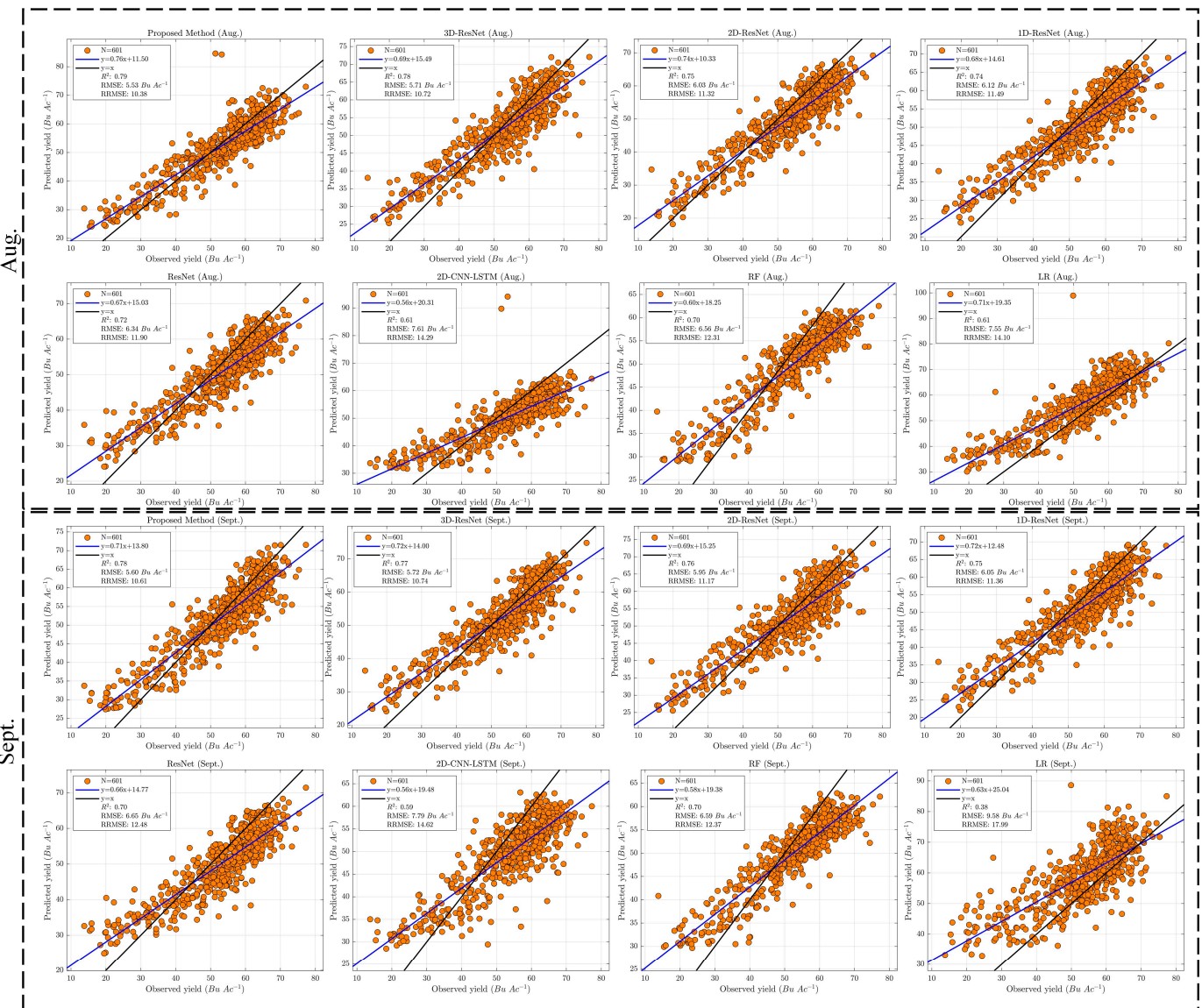

**Figure 8.** Scatter plots of growing in-season predicted vs. USDA yield using the proposed method and compared methods in 2021.

It is important to note that our analysis primarily focused on lower-level features from Sentinel-1 images, which may have influenced the results. Additionally, the quantity of the data used in the analysis can also affect the model's performance. Nonetheless, these values confirm the robust performance and validity of the proposed method throughout the soybean-growing season.

The $R^2$ values of the proposed method reached 0.794 and 0.788 in August and September, respectively. In Figure 8, when using 3D-ResNet-BiLSTM, the fit line (depicted in blue, representing the regression line between the predicted and observed yield values) is closely aligned with the diagonal line (shown in black, signifying perfect agreement between the predicted and actual yield values), and predictions were clustered reasonably around the diagonal line. This proximity to the diagonal line indicates a stronger correlation between the predicted and actual yield values when using the proposed method. Additionally, the 1D/2D/3D-ResNet and ResNet models demonstrated good agreement between the predicted and observed yield values.

Figure 9 presented a visual depiction highlighting the distribution of soybean yield by comparing the USDA yield with the predicted yield derived from the proposed method. The results in Figure 9 demonstrated a substantial agreement between the observed and predicted soybean yield during the analysis, reinforcing the reliability and accuracy of our proposed method's predictions.

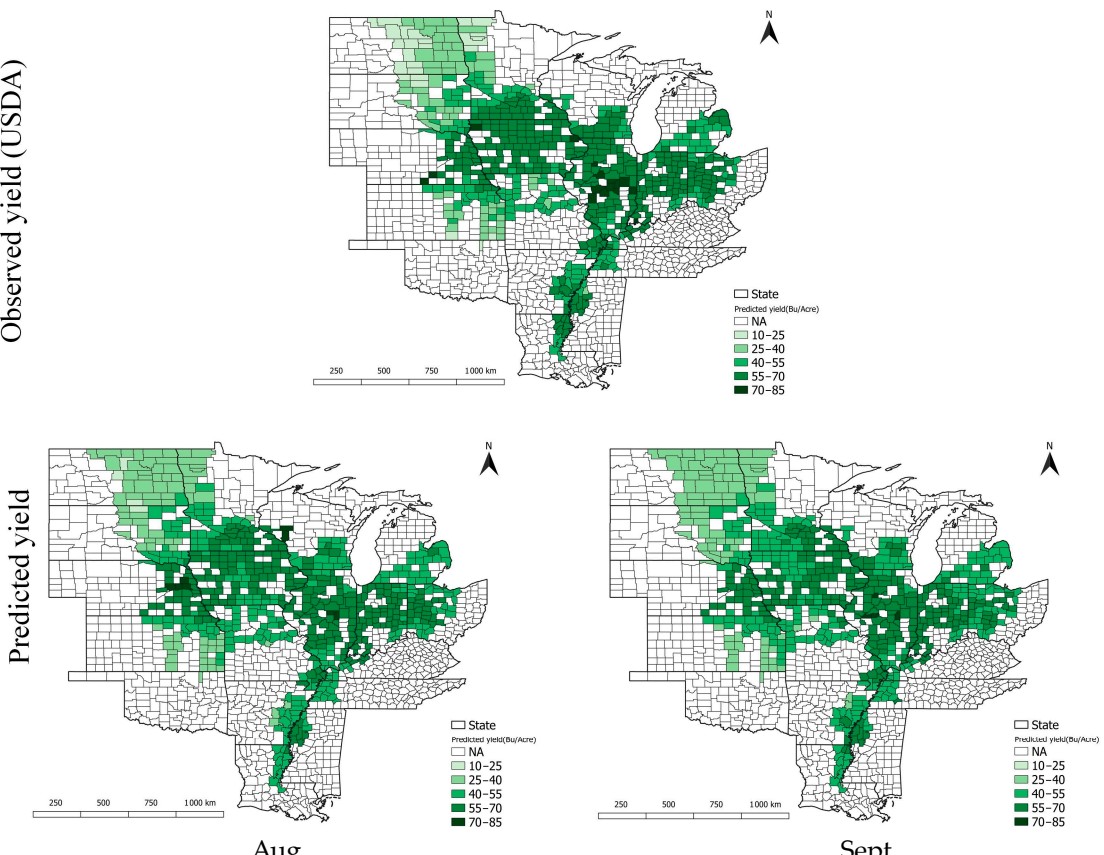

**Figure 9.** Map of USDA soybean yield and predicted soybean yield in 2021.

Based on the USDA yield map presented in Figure 9, it was evident that counties in states such as North Dakota, South Dakota, Kansas, Missouri, Minnesota, and Oklahoma experienced comparatively lower yields in 2021. In contrast, counties like Iowa, Nebraska, Illinois, Indiana, Ohio, Kentucky, Tennessee, Arkansas, Mississippi, Louisiana, Michigan, and Wisconsin displayed higher yields during the same period.

Figure 10 depicts the average accuracy of our proposed method compared to other models for August and September. The average $R^2$ values for the 3D-ResNet-BiLSTM, 3D-ResNet, 2D-ResNet, 1D-ResNet, ResNet, 2D-CNN-LSTM, RF, and LR models were 0.791, 0.779, 0.758, 0.716, 0.60, 0.708, and 0.499, respectively.

In Figure 10, our investigation shows significant performance improvements achieved via various architectural modifications. Firstly, the inclusion of Conv1D demonstrated a noteworthy 3.49% improvement in the performance of the ResNet model. Secondly, incorporating Conv2D contributed a substantial 4.28% improvement in ResNet's performance. Thirdly, the adoption of Conv3D proved highly effective, resulting in an impressive 6.41% improvement in ResNet's performance. Lastly, adding the BiLSTM layer enhanced the performance of the 3D-ResNet model by a commendable 1.12%.

Moreover, our proposed model utilizing the Sentinel 1–2 and Daymet data has demonstrated a significant increase of 19.025% in accuracy for soybean yield prediction compared to the 2D-CNN-LSTM model presented by Sun et al. [5]. Error maps indicate that certain cities exhibit the lowest errors in August, while others show the weakest errors in Septem-

ber, indicating the time difference between sowing and harvesting. Our proposed method achieves a high accuracy with an $R^2$ of 0.791 in August and September. Table 5 presents the MAPE metric for each state using the 3D-ResNet-BiLSTM model.

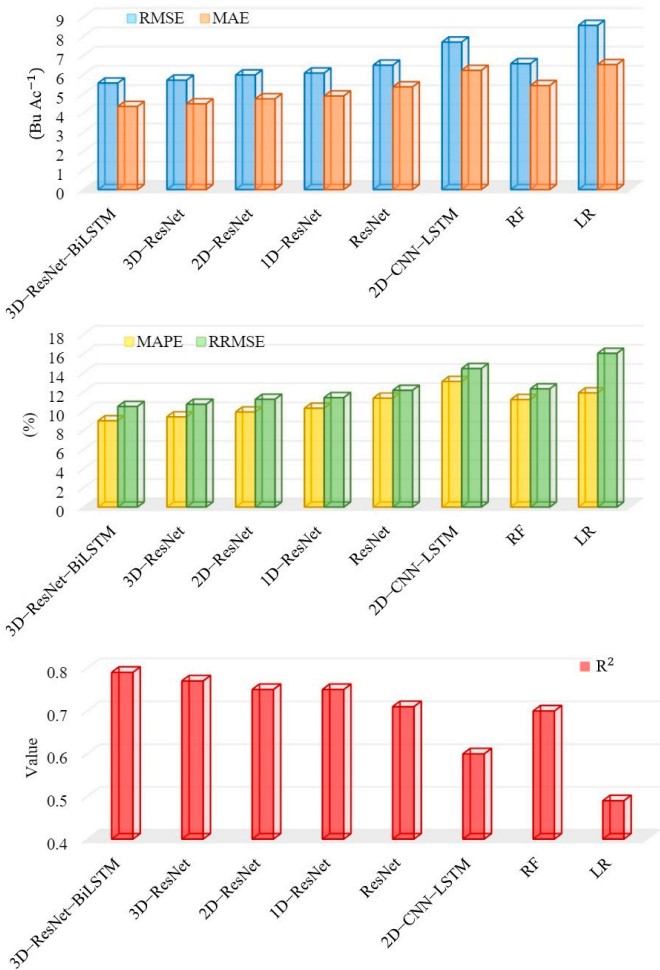

**Figure 10.** The average accuracy of our proposed method and considered models between August and September.

**Table 5.** The evaluation of soybean yield prediction in each U.S. state was conducted based on the proposed 3D-ResNet-BiLSTM model.

| U.S. State | RMSE (Bu Ac$^{-1}$) | MAE (Bu Ac$^{-1}$) | MAPE (%) | RRMSE (%) |
|---|---|---|---|---|
| Arkansas | 6.74 | 5.60 | 11.08 | 13.00 |
| Illinois | 5.17 | 4.17 | 6.57 | 8.17 |
| Indiana | 4.53 | 3.49 | 5.74 | 7.55 |
| Iowa | 5.97 | 4.81 | 7.62 | 9.59 |
| Kansas | 5.54 | 4.75 | 13.25 | 13.75 |
| Kentucky | 4.50 | 3.75 | 6.62 | 7.92 |
| Louisiana | 6.94 | 6.05 | 10.75 | 12.62 |
| Michigan | 3.62 | 2.86 | 5.75 | 7.07 |
| Minnesota | 5.57 | 4.38 | 11.36 | 11.30 |
| Mississippi | 5.01 | 3.83 | 7.54 | 9.06 |

**Table 5.** *Cont.*

| U.S. State | RMSE (Bu Ac$^{-1}$) | MAE (Bu Ac$^{-1}$) | MAPE (%) | RRMSE (%) |
|---|---|---|---|---|
| Missouri | 5.12 | 4.08 | 9.07 | 10.61 |
| Nebraska | 5.72 | 4.62 | 7.43 | 9.32 |
| North Dakota | 6.54 | 5.40 | 25.09 | 25.01 |
| Ohio | 4.08 | 3.46 | 5.94 | 7.13 |
| Oklahoma | 18.29 | 18.29 | 132.51 | 132.51 |
| South Dakota | 4.21 | 3.60 | 9.98 | 10.64 |
| Tennessee | 4.41 | 3.36 | 6.48 | 8.65 |
| Wisconsin | 8.74 | 6.22 | 11.16 | 15.53 |

After comparing our proposed method with other models, it became evident that the 1D/2D/3D-ResNet models consistently outperformed the ResNet, 2D-CNN-LSTM, RF, and LR models. Furthermore, we observed that the ResNet model's yield prediction accuracy improved notably when employing Conv3D layers instead of Conv1D/2D and dense layers. In stark contrast, the Linear Regression model exhibited the poorest performance among all the evaluated models.

## 4. Discussion

This study introduced the 3D-ResNet-BiLSTM model as a new predictor for forecasting county-level soybean yield using a combination of Sentinel-1 and Sentinel-2 imagery and Daymet climate data. Unlike widely-used approaches [5,9,16,17,19,21] that rely on MODIS products, which are limited by their coarse spatial resolution, our study demonstrates the value of integrating medium-resolution Sentinel 1–2 data with climate data for developing more accurate yield prediction models. Additionally, we achieved improved performance of the 3D-ResNet-BiLSTM model by significantly reducing the input tensor size by a factor of 57.81 compared to MODIS data [5], facilitating early soybean yield predictions and boosting the efficiency of the model training process.

Our study also examined the sensitivity of network architecture complexity in predicting soybean yield, particularly in scenarios with short-period training data. While previous research by Sun et al. [5] has predominantly used 2D-CNN-LSTM architectures for soybean yield prediction, these architectures often encounter an ill-posed problem when confronted with insufficient/short-period training data due to more unknown parameters. Our results demonstrate the capability of the proposed 3D-ResNet-BiLSTM architecture to handle situations with limited/short-period training data effectively.

Furthermore, our research highlights the substantial advantages of combining feature extraction with the ResNet and yield prediction with BiLSTM, leveraging the satisfactory spatial resolution of Sentinel-1 and Sentinel-2 imagery to achieve accurate predictions of soybean yield at the county level. Additionally, our implementation of three CNN models—Conv1D, Conv2D, and Conv3D—revealed that using Conv3D significantly minimized the MAPE than Conv2D and Conv1D (see Table 4). This superior performance of Conv3D can be attributed to its capacity to extract spatial and temporal information from time-series data [23].

While our study demonstrates the effectiveness of the 3D-ResNet-BiLSTM model for soybean yield prediction, further research is needed to fully validate its performance across different geographical regions and under diverse environmental conditions. Additionally, exploring the integration of additional data sources, such as soil data or agricultural management practices, could further enhance the accuracy and generalizability of the model.

## 5. Conclusions

Soybean, a crucial commodity in U.S. agriculture, demands accurate regional yield forecasting for informed planning decisions. This study harnesses diverse data sources, including Sentinel-1, Sentinel-2, and Daymet data, extracting a comprehensive set of 19 features encompassing spectral bands, vegetation indices, SAR polarizations, and critical weather parameters. A novel 3D-ResNet architecture was designed to process these diverse inputs effectively, featuring a unique combination of 3D convolutional and recurrent layers. This architecture extracts high-level features subsequently fed into a BiLSTM layer, enabling precise prediction of soybean yield. To assess the efficacy of our model, we trained it on data from 2019 to 2020 and set its performance using data from 2021. Evaluating the proposed 3D-ResNet-BiLSTM model against other models revealed its remarkable performance, achieving an $R^2$ of 0.79 and an RMSE of 5.56 Bu Ac-1, surpassing all other considered models by a significant margin. This significant improvement can be mainly due to the model's capacity to effectively capture spatial and temporal patterns in the data, a crucial aspect for accurate yield prediction in areas with complex terrain and variable weather patterns. These findings underscore the transformative potential of fusing advanced RS data, feature-rich datasets, and state-of-the-art deep learning models to pave the way for data-driven agricultural decision-making. This approach not only enhances yield forecasting accuracy but also holds promise for optimizing resource allocation, improving crop management practices, and ultimately strengthening food security. As we move forward, integrating additional data sources, such as soil data and agricultural management practices, can further enhance the accuracy and generalizability of these models, leading to even more informed and sustainable farming practices.

**Author Contributions:** Conceptualization, M.F., R.S.-H. and A.M.; Methodology, M.F., R.S.-H. and A.M.; Project administration, R.S.-H.; Resources, M.F., R.S.-H. and A.M.; Validation, M.F., R.S.-H. and A.M.; Supervision, R.S.-H.; Writing—original draft, M.F., R.S.-H. and A.M.; Writing—review and editing, M.F., R.S.-H. and A.M. All authors have read and agreed to the published version of the manuscript.

**Funding:** This research received no external funding.

**Data Availability Statement:** The data that support the findings of this study are available on reasonable request from the corresponding author.

**Acknowledgments:** We would like to express our sincere gratitude to Google for providing access to Earth Engine and Colab. These pivotal tools greatly facilitated the execution and analysis of this research. Additionally, we extend our heartfelt thanks to the United States Department of Agriculture (USDA) for generously providing the essential yield values that contributed significantly to the success of this study.

**Conflicts of Interest:** The authors declare no conflict of interest.

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
