# Peer review of "3D-ResNet-BiLSTM Model: A Deep Learning Model for County-Level Soybean Yield Prediction with Time-Series Sentinel-1, Sentinel-2 Imagery, and Daymet Data"

_remotesensing, doi:10.3390/rs15235551_

Round 1

Reviewer 1 Report

Comments and Suggestions for Authors

One of the main tasks of agriculture and crop production is to ensure sufficient production of high-quality and healthy food resources for human and animal nutrition. Soybean is a globally important crop that is a source of high-quality protein. I consider field crop yield forecasting, including soybeans, based on various remote sensing algorithms to be an important area of ​​research. The scientific article has the required quality and suitable structure. In the conclusions, the authors indicated that using extensive dataset, introduced the innovative 3D-ResNet-BiLSTM model to efficiently predict soybean yield by limited training dataset. The results indicated that the features extracted from high spatial resolution Sentinel 1-2 images were superior in accurately predicting soybean yield. Forty literary sources were cited in the manuscript, which testifies to a good professional review of the authors. I did not find any serious defects in the work or in the presentation or ethical problems. In my opinion, the keywords are consistent with the content of the article. I do not have any serious comments and manuscript „3D-Resnet-BiLSTM Model: A Deep Learning Model for County-Level Soybean Yield Prediction with Time Series Sentinel-1, Sentinel-2 Imagery, and Daymet Data “ meets the requirements and can be published in a Journal Remote Sensing.

Author Response

Response to Reviewer 1:

Response: Thank you for your very insightful and valuable comments and recommendations. They helped us greatly to improve the paper. We have made further modifications to the revised manuscript according to the reviewer's suggestion. The detailed corrections are highlighted.

Reviewer comments: 1. One of the main tasks of agriculture and crop production is to ensure sufficient production of high-quality and healthy food resources for human and animal nutrition. Soybean is a globally important crop that is a source of high-quality protein. I consider field crop yield forecasting, including soybeans, based on various remote sensing algorithms to be an important area of ​​research. The scientific article has the required quality and suitable structure. In the conclusions, the authors indicated that using an extensive dataset, introduced the innovative 3D-ResNet-BiLSTM model to efficiently predict soybean yield by limited training dataset. The results indicated that the features extracted from high spatial resolution Sentinel 1-2 images were superior in accurately predicting soybean yield. Forty literary sources were cited in the manuscript, which testifies to a good professional review of the authors. I did not find any serious defects in the work or in the presentation or ethical problems. In my opinion, the keywords are consistent with the content of the article. I do not have any serious comments and the manuscript "3D-Resnet-BiLSTM Model: A Deep Learning Model for County-Level Soybean Yield Prediction with Time Series Sentinel-1 Sentinel-2 Imagery, and Daymet Data" meets the requirements and can be published in a Journal Remote Sensing.

Response to comments 1: We appreciate your valuable suggestions and thank you for recommending relevant papers that align with the scope of our study. Thank you once again for your time and effort in reviewing my work.

Reviewer 2 Report

Comments and Suggestions for Authors

1. There are formatting issues in the manuscript, including the numerical formatting of the title and the spelling of line 39, as well as some errors in the following sections. In addition, many images have low resolution, making it difficult to see the legends in the images clearly.

2. The manuscript lacks innovation. The author states that the modis data resolution is insufficient, while Sentinel-2 has high resolution and has received less attention. But currently, most crop yield estimation articles use sentinel-2 data.

3. The author states that most estimation models are relatively robust under sufficient data conditions, while there are few experiments in the absence of data. But according to the author's subsequent experiments, it can be seen that the experiment designed by the author used Sentinel 1/2 data and climate data for more than 8 months. Therefore, the data is already relatively sufficient. We hope the author provides a clear definition of the adequacy of data. In the process of designing the experiment, the author needs to express why the data in the manuscript is insufficient.

4. The author only connects 3D-ResNet and bidirectional LSTM together. We think that the innovation of the network is insufficient, and we hope that the author can clearly state the purpose of designing the network in the manuscript.

Lines 299-307:

The training loss of 2D-CNN-LSTM is lower than the validation loss, which the author believes is underfitting. However, according to Table 3, this network has the largest number of parameters and generally has strong fitting ability. I hope the author can give an explanation. I believe that the experiment the author designed was a case of insufficient data, more likely due to overfitting but underfitting. I also wanted the author to provide an explanation.

Comments on the Quality of English Language

Need extensive English editing

Author Response

Response to Reviewer 2:

Response: Thank you for your very insightful and valuable comments and recommendations. They helped us greatly to improve the paper. We have made further modifications to the revised manuscript according to the reviewer's suggestion. The detailed corrections are highlighted in Cyan.

Reviewer comments: 1. There are formatting issues in the manuscript, including the numerical formatting of the title and the spelling of line 39, as well as some errors in the following sections. In addition, many images have low resolution, making it difficult to see the legends in the images clearly.

Response to comments 1: Thank you for bringing these formatting issues to our attention. We carefully reviewed the manuscript and corrected the numerical formatting and spelling errors, including line 39. For example, a spelling correction in line 39 is the wording in the following phrase: "Having a high oil and protein content makes soybeans a vital crop for food security, and the United States (U.S.) is the top global producer of this valuable commodity." We also improved the resolution of all images from Figure 1 to Figure 10 to make the legends visible. Your feedback is valuable, and we appreciate your attention to detail.

Reviewer comments: 2. The manuscript lacks innovation. The author states that the MODIS data resolution is insufficient, while Sentinel-2 has high resolution and has received less attention. But currently, most crop yield estimation articles use sentinel-2 data.

Response to comments 2: Thank you for your comment. Our study focused on soybean crop yield estimation in the U.S., primarily investigated using MODIS images due to high temporal resolution. To clarify our sentence, we have rewritten it as follows:

The MODIS has been extensively employed for soybean yield prediction due to its high temporal resolution, but its accuracy is limited by its low spatial resolution. However, the potential of even higher-resolution images, such as those from Sentinel-2, which provide rich spectral information, including red-edge bands, needs more attention. Additionally, the potential of combining Sentinel-2 and Sentinel-1 images, along with weather and climatology variables, to improve prediction accuracy has been less regarded.”

Reviewer comments: 3. The author states that most estimation models are relatively robust under sufficient data conditions, while there are few experiments in the absence of data. But according to the author's subsequent experiments, it can be seen that the experiment designed by the author used Sentinel 1/2 data and climate data for more than 8 months. Therefore, the data is already relatively sufficient. We hope the author provides a clear definition of the adequacy of data. In the process of designing the experiment, the author needs to express why the data in the manuscript is insufficient.

Response to comments 3: Thank you for the feedback, reviewer; as mentioned in the introduction, most researchers have mainly used long time span datasets (periods over five years) to train deep learning models (Khaki et al., 2020; Sun et al., 2019; Zhu et al., 2022). While these models have demonstrated robust prediction abilities with long-time span training data, assessing their performance in scenarios with limited data is imperative. Moreover, most researchers used CNN-LSTM-based models for soybean yield estimation. Considering that 2D-CNN-LSTM architectures have a more significant number of weight and bias parameters, they require a significant amount of training data to estimate these parameters. Therefore, one of our goals was to design a network with an optimal number of parameters and high accuracy against short-time span training data. We clarified the novelties of our work in the last paragraph of the introduction.

Reviewer Comments: 4. The author only connects 3D-ResNet and bidirectional LSTM together. We think that the innovation of the network is insufficient, and we hope that the author can clearly state the purpose of designing the network in the manuscript.

Response to comments 4: Thank you for the reviewer’s feedback. We understand your concerns regarding the lack of innovation in our network design and would like to address this issue. According to studies, there are three different approaches to developing deep learning architectures: 1) designing a new architecture based on how input data is fed to the network, 2) connecting different networks, and 3) a combination of the first and second approaches. Our research has revealed that various studies have been conducted in the field of CNN and RNNs, including 3D-CNN, LSTM, and ResNet16 (Manoj et al., 2022; Schwalbert et al., 2020; Terliksiz & Altýlar, 2019). Also, different architectures have been presented by connecting the CNN family and the RNN family, such as 2D-CNN-LSTM, 3D-CNN-attention-LSTM, and CNN-RNN (Khaki et al., 2020; Nejad et al., 2023; Sun et al., 2019). Additionally, we found that LSTM, a family of RNNs, has garnered more interest than Bidirectional RNNs in the field of crop estimation. Bidirectional RNNs are effective in cases where a full understanding of temporal information is needed, as they look at both sides of the sequence (forward-backward and backward-forward), allowing the network to use past and future information for any point in time (Siami-Namini et al., 2019).

ResNet50 has the advantage of effectively training very deep neural networks with a large number of layers. The use of residual connections in ResNet50 helps alleviate the problem of vanishing gradients, which can occur in deep networks and make training difficult. This enables the successful training of deeper networks and leads to improved performance in tasks such as image classification and prediction (Chen et al., 2020).

The Conv3D (3D Convolutional Neural Network) is particularly useful for capturing spatial and temporal features in 3D data, making it effective for tasks such as classification and prediction (Rao & Liu, 2020).

Based on our research, we have determined that a network utilizing the combined potential of 3D-CNN, ResNet-50, and Bidirectional LSTMs is needed in the field of crop estimation. To address this, we have considered ResNet-50 as our base model. By embedding the Conv with different dimensions (1D, 2D, and 3D) in the ResNet-50 network, we have found that 3D-ResNet can learn spatial and temporal features more effectively than ResNet-50 alone. Additionally, the use of 3D convolutional layers in the ResNet-50 component helps reduce computational cost and memory requirements compared to traditional 3D convolutional networks, making it more efficient for 3D data processing.

After extracting spatial and temporal features with the help of 3D-ResNet, these features were entered into the Bi-LSTM layer to extract complex temporal information fully. Therefore, our proposed architecture has three main advantages: solving the vanishing gradient problem with the help of ResNet-50, extracting spatial and temporal features (together) with 3D-ResNet, and fully extracting complex temporal information with the help of Bi-LSTM.

Thanks to the respected reviewer, as explained in the answer to comment 2, the 2D-CNN-LSTM architecture designed by Sun et al. had a high number of parameters compared to our proposed method, which required 10-year time periods to solve these parameters. Our goal was to provide a network that provides acceptable results for 3-year periods (2 years of training data and one year of test data). The results presented in Section 3.2, under 'Comparative results of soybean yield prediction' in Table 4, demonstrate that our proposed network yields more favorable results compared to 2D-CNN-LSTMs for shorter training periods. On the other hand, the model with the most parameters will not always have high accuracy, and achieving high accuracy requires a network with an optimal architecture and the proportion of the weight and bias parameters with the number of training data.

Reviewer comments: 5: Lines 299-30:The training loss of 2D-CNN-LSTM is lower than the validation loss, which the author believes is underfitting. However, Table 3 shows that this network has the most significant number of parameters and generally has strong fitting ability. I hope the author can give an explanation. I believe that the experiment the author designed was a case of insufficient data, more likely due to overfitting but underfitting. I also wanted the author to provide an explanation.

Response to comments 5: Thanks for the valuable feedback. We have made corrections to the analysis pertaining to lines 299-308. The following results have been derived based on the validation loss curves depicted in Figure 5. The validation loss curves indicate that there were higher fluctuations in August than in September. These curves also demonstrate that the addition of extracted features in subsequent months led to a reduction in the range of fluctuations. When significant fluctuations or erratic behavior in validation loss are observed, particularly when the training loss is consistently decreasing, it may suggest that the model is overfitting the training data (Zhang et al., 2019).

The analysis of Loss-Validation curves for 2D-CNN-LSTM did not indicate overfitting or under-fitting, and a slight improvement in results could be achieved by increasing the number of epochs. However, in the Loss-Validation curve for 2D-ResNet in September, overfitting was observed between epochs 800 and 1000, as evidenced by a substantial increase in the gap between Loss-Validation and Loss-Training. This could be attributed to significant disparities between the batches in the validation and training data (Zhang et al., 2019).

In general, for yield prediction, it may be more appropriate to save the best model based on loss validation, as there is a possibility that the model may not achieve proper convergence with the inclusion of validation data. Consequently, we have saved the best model among all the trained models. It is important to note that the discussion of convergence analysis of training models also hinges on the error calculation criteria and test phase results (Zhang et al., 2019).

The diagram in Figure 10 illustrates that our proposed model achieved an acceptable R2 of 79.10% and MAPE of 9% on the test data, delivering satisfactory outcomes.

Reviewer Comments: 6: Need extensive English editing

Response to comments 6: Thank you for your suggestion. The manuscript has been proofread and corrected by a native speaker to ensure extensive English editing.

Reference

Chen, D., Hu, F., Nian, G., & Yang, T. (2020). Deep residual learning for nonlinear regression. Entropy, 22(2), 193.

Khaki, S., Wang, L., & Archontoulis, S. V. (2020). A cnn-rnn framework for crop yield prediction. Frontiers in Plant Science, 10, 1750.

Manoj, T., Makkithaya, K., & Narendra, V. G. (2022). A Federated Learning-Based Crop Yield Prediction for Agricultural Production Risk Management. 2022 IEEE Delhi Section Conference, DELCON 2022. https://doi.org/10.1109/DELCON54057.2022.9752836

Nejad, S. M. M., Abbasi-Moghadam, D., Sharifi, A., Farmonov, N., Amankulova, K., & Laszlz, M. (2023). Multispectral Crop Yield Prediction Using 3D-Convolutional Neural Networks and Attention Convolutional LSTM Approaches. IEEE Journal of Selected Topics in Applied Earth Observations and Remote Sensing, 16. https://doi.org/10.1109/JSTARS.2022.3223423

Rao, C., & Liu, Y. (2020). Three-dimensional convolutional neural network (3D-CNN) for heterogeneous material homogenization. Computational Materials Science, 184. https://doi.org/10.1016/j.commatsci.2020.109850

Schwalbert, R. A., Amado, T., Corassa, G., Pott, L. P., Prasad, P. V. V., & Ciampitti, I. A. (2020). Satellite-based soybean yield forecast: Integrating machine learning and weather data for improving crop yield prediction in southern Brazil: Agricultural and Forest Meteorology, 284, 107886.

Siami-Namini, S., Tavakoli, N., & Namin, A. S. (2019). The performance of LSTM and BiLSTM in forecasting time series. 2019 IEEE International Conference on Big Data (Big Data), 3285–3292.

Sun, J., Di, L., Sun, Z., Shen, Y., & Lai, Z. (2019). County-level soybean yield prediction using deep CNN-LSTM model. Sensors (Switzerland), 19(20). https://doi.org/10.3390/s19204363

Terliksiz, A. S., & Altýlar, D. T. (2019). Use of deep neural networks for crop yield prediction: A case study of soybean yield in lauderdale county, alabama, usa. 2019 8th International Conference on Agro-Geoinformatics (Agro-Geoinformatics), 1–4.

Uribeetxebarria, A., Castellón, A., & Aizpurua, A. (2023). Optimizing Wheat Yield Prediction Integrating Data from Sentinel-1 and Sentinel-2 with CatBoost Algorithm. Remote Sensing, 15(6). https://doi.org/10.3390/rs15061640

You, J., Li, X., Low, M., Lobell, D., & Ermon, S. (2017). Deep gaussian process for crop yield prediction based on remote sensing data. Proceedings of the AAAI Conference on Artificial Intelligence, 31(1).

Zhang, H., Zhang, L., & Jiang, Y. (2019). Overfitting and Underfitting Analysis for Deep Learning Based End-to-end Communication Systems. 2019 11th International Conference on Wireless Communications and Signal Processing, WCSP 2019. https://doi.org/10.1109/WCSP.2019.8927876

Zhu, Y., Wu, S., Qin, M., Fu, Z., Gao, Y., Wang, Y., & Du, Z. (2022). A deep learning crop model for adaptive yield estimation in large areas. International Journal of Applied Earth Observation and Geoinformation, 110, 102828.

Reviewer 3 Report

Comments and Suggestions for Authors

This manuscript aims to predict soybean yield with time series Sentinel-1, Sentinel-2 imagery, and Daymet data. A 3D-Resnet-BiLSTM model was proposed to predict soybean yield at the country level across the U.S., even with a restricted number of RS training samples. The new model achieved a 7% higher R2 compared to ResNet and RF models, and outperformed LR and 2D-CNN-LSTM models. The most concern of this paper is the structure, especially for the Result and Discussion sections. However, there are some revisions that should be addressed before the publication.

1. In Lines 29 and 30, the R2 and R2 should be R2.

2. What means the Band 1cdl(Gray) in Fig.1?

3. Fig. 2 shows the flowchart of this study, and the irrelevant information should be deleted.

4. Fig. 3 are showing the result, which should not be in the Materials and Method section.

5. The monthly composites are maximum or median value? Any others? Which algorithm was used for making cloudy? For Sentinel-1, there is no need to do this.

6. The Lines 279-294 of Result parts, these sentences are described how the study was done? But not the result.

7. In line 393, what is the compared to the 23-CNN-LSTM model presented by?. ?

8. Some decimal places have two digits and some places have three or four digits.

9. In lines 412 and 418, the MADIS data is MODIS?

10. The high-level and lower-level features refer to ? And how to determine? In other word, are these feature determined by their contribution to yield predicting.

11. As mentioned in the introduction, temperature is also important in yield prediction, but it is not used in this study. Why ?

Author Response

Response to Reviewer 3:

Response: Thank you for your very insightful and valuable comments and recommendations. They helped us greatly to improve the paper. We have made further modifications to the revised manuscript according to the reviewer's suggestion. The detailed corrections are highlighted in magenta.

Reviewer comments. This manuscript aims to predict soybean yield with time series Sentinel-1, Sentinel-2 imagery, and Daymet data. A 3D-Resnet-BiLSTM model was proposed to predict soybean yield at the country level across the U.S., even with a restricted number of RS training samples. The new model achieved a 7% higher R2 compared to ResNet and RF models, and outperformed LR and 2D-CNN-LSTM models. The most concern of this paper is the structure, especially for the Result and Discussion sections. However, there are some revisions that should be addressed before the publication.

Reviewer comments: 1. In Lines 29 and 30, the 'R2'and 'R2' should be R2.

Response to comments 1: Thank you very much for your valuable comments; we genuinely appreciate the time and effort you invested in reviewing our article and providing valuable feedback. We applied formatting, and structure corrections to all sections. We applied spelling corrections, including spelling corrections on line 30 'Furthermore, the 3D-Resnet-BiLSTM model showed a 7% higher R2 compared to ResNet and RF models, and enhanced by 27% and 17% against LR and 2D-CNN-LSTM models, respectively.'

Reviewer comments: 2. What means the 'Band 1cdl (Gray)' in Fig.1?

Response to comments 2: Thank you for your insightful comment. Figure 1 shows the spatial distribution of soybean fields in the United States, shown in green. Therefore, in Figure 1, in the legend, we replaced 'Band 1cdl (Gray)' with 'Spatial distribution of soybean fields'

Reviewer comments: 3.  Fig. 2 shows the flowchart of this study, and the irrelevant information should be deleted.

Response to comments 3: Thanks for your feedback; according to the reviewer, we deleted irrelevant and additional information from the flowchart in Figure 2.

Reviewer comments: 4.   Fig. 3 are showing the result, which should not be in the Materials and Method section.

Response to comments 4: Thanks for your feedback, Figure 3 shows the average time series curves of the generated monthly composites from extracted features from Sentinel 1-2 and Daymet data for training data, which are presented in the dataset just for information of reader.

Reviewer comments: 5. The monthly composites are maximum or median value? Any others? Which algorithm was used for making cloudy? For Sentinel-1, there is no need to do this.

Response to comments 5: Thank you for your valuable comments; we used the median value to generate monthly composites. The median value is the middle value in a set of numbers arranged in ascending order. The median is the middle number if the set has an odd number of values. If the set has an even number of values, the median is the average of the two middle numbers. In image compositing, the median value is used to calculate the final pixel value by taking the median of corresponding pixel values from multiple original images. The advantage of using the median value in image compositing is that it helps to reduce the impact of noise, outliers, and extreme values in the original images. Overall, using the median value in image compositing can help to produce high-quality and natural-looking composite images.

Sentinel-1 is a radar imaging satellite capable of penetrating clouds and capturing images of the Earth's surface. It can provide valuable data for creating composite images free from cloud cover. Therefore, there is no need to remove the cloud.

For Sentinel 2, an optical image, the 'QA60' band contains information on whether the pixel is cloudy in the 10th and 11th bit for opaque and cirrus clouds. So, in the Google Earth Engine system, we can check that by checking values that have 1 on the 10th and 11th bit, or we can use 'bitwiseAnd' to achieve the same.  Thus, we applied 'qa.bitwiseAnd' to mask clouds using the Sentinel-2 'QA60' band in google earth engine (https://developers.google.com/earth-engine/datasets/catalog/COPERNICUS_S2_SR_HARMONIZED).

Reviewer comments: 6. The Lines 279-294 of Result parts, these sentences are described how the study was done? But not the result.

Response to comments 6: Thanks for your feedback; you are correct that the sentences in lines 279-294 of the '3.1 Experimental setup' section do not directly present the study results. Instead, they describe the experimental setup or how the study was conducted. The '3.1 Experimental setup' section aims to provide readers with an understanding of the study's experimental design and setup. The actual results of the study were presented in '3.2. Comparative results of the soybean yield prediction sections. According to the opinion of the honorable reviewer, we added our results in the continuation of lines 279-294.

Reviewer comments: 7. In line 393, what is the “compared to the 2-CNN-LSTM” model presented by?.” ?

Response to comments 7: Thanks for your feedback, a correction in line 393 is that the wording in the following phrase  'Moreover, our proposed model utilizing Sentinel 1-2 and Daymet data has demonstrated a significant increase of 19.025% in accuracy for soybean yield prediction com-pared to the 2D-CNN-LSTM model presented by Sun et al, [5]'.

Reviewer comments: 8.. Some decimal places have two digits and some places have three or four digits.

Response to comments 8: Thank you for bringing this to our attention. We apologize for the inconsistency in the number of decimal places. We checked the number of decimal places and corrected throughout the document.

Reviewer comments: 9.. In lines 412 and 418, the MADIS data is MODIS?

Response to comments 9: Thank you for bringing this to our attention. A correction in line 412-418 is that the wording in the following phrase  ' Our primary focus centered on analyzing the sensitivity of network architecture complexity in predicting soybean yield, particularly in scenarios with limited training data. Previous research predominantly utilized 2D-CNN, LSTM, and 2D-CNN-LSTM architectures to predict soybean yield using MODIS data.'

Reviewer comments: 10.  The high-level and lower-level features refer to ? And how to determine? In other word, are these feature determined by their contribution to yield predicting.

Response to comments 10: Thank you for your insightful comments. Deep learning models are designed to learn and extract features from input data at various levels of abstraction. In the initial layers of the model, they capture low-level features such as edges, textures, and basic shapes. As the data passes through the deeper layers of the model, it extracts higher-level features that represent more complex patterns and relationships in the data. Therefore, in deep learning, low-level features refer to basic features while high-level features refer to more complex features. In deep learning, input data can be considered low-level features because it contains raw information about the input samples. The deep learning model then learns the high-level features through multiple layers of processing and feature extraction. Thus, deep learning models capture low-level features across the initial layers and high-level features in the deeper layers (https://www.researchgate.net/publication/249862831_INTERACTION_BETWEEN_MODULES_IN_LEARNING_SYSTEMS_FOR_VISION_APPLICATIONS?enrichId=rgreq-2f6f1c3f9d3eeafa23aa3f1ffd6b94b2-XXX&enrichSource=Y292ZXJQYWdlOzI0OTg2MjgzMTtBUzoxMDM0OTE0ODE2MzY4NjVAMTQwMTY4NTcwMDE0NA%3D%3D&el=1_x_2&_esc=publicationCoverPdf).

Reviewer comments: 11. As mentioned in the introduction, temperature is also important in yield prediction, but it is not used in this study. Why ?

Response to comments 11: Thanks to the valuable comment of the respected reviewer, in this research we used Sentinel 1-2 images with high spatial resolution. Because Sentinel 2 images do not have a thermal band, we were not able to calculate the temperature of the Land surface. Therefore, based on the suggestion of the honorable reviewer, in future research, we intend to work in a multi-sensor approach and evaluate the impact of other bands of Daymet data, including evapotranspiration, soil moisture, and Palmer Drought Severity Index, along with soil characteristics, including soil organic carbon content, cation exchange, Organic carbon stocks, etc.

Round 2

Reviewer 2 Report

Comments and Suggestions for Authors

The authors addressed all my comments and made the necessary changes. Manuscripts can be published in current form

Comments on the Quality of English Language

Minor editing required for English

Reviewer 3 Report

Comments and Suggestions for Authors

 The authors have done a wonderful job of making revisions according to the comments. Particularly, the figures are much improved and easier to read. However, the manuscript will need some minor revisions before it can be accepted for publication.

1. In lines 30-31, the 'R2' should be changed to R2. In addition, the “an R2 of 79%” should be an R2 of 0.79 in line 315. Please, check carefully through the manuscript, such as Table 4, line 350 and line 376.

2. The decimal place in this manuscript are two digits expect for Table 6. Please change the Table 6.